# Pharmacist Involvement in Population Health Management for a Pediatric Managed Medicaid Accountable Care Organization

**DOI:** 10.3390/children6070082

**Published:** 2019-07-04

**Authors:** Catherine Kuhn, Brigid K. Groves, Chester Kaczor, Sonya Sebastian, Ujjwal Ramtekkar, Joshua Nowack, Christina Toth, Olivia Valenti, Charitha Gowda

**Affiliations:** 1Updox, 6555 Longshore Street, Dublin, OH 43017, USA; 2Partners For Kids, 700 Children’s Drive, Columbus, OH 43205, USA; 3Nationwide Children’s Hospital, 700 Children’s Drive, Columbus, OH 43205, USA

**Keywords:** pharmacist, population health management, medication management, accountable care organization, quality improvement

## Abstract

Accountable care organizations (ACOs) have emerged as an effective healthcare delivery model for managing quality and cost at a population level. Within ACOs, pharmacists are critical for the delivery of high-value health care, offering patients and health care providers medication-related training, resources, and guidance that can improve quality of care at lower costs. Partners For Kids (PFK), one of the oldest and largest pediatric ACOs in the country, has successfully leveraged pharmacists to provide population health management and medication management to promote health outcomes for individual patients and the overall population it serves. This review explores how the inclusion of pharmacists in the development and execution of various quality improvement initiatives within PFK has positively impacted outcomes for patients while also lowering overall spend. A catalog of interventions is provided to offer various ways that pharmacists can intersect as providers in the triad of patient/family, payor, and provider. By providing enhanced training and education, on-site guidance, medication management, and population-level data analysis, pharmacists are able to identify and improve inefficiencies in care. Moving forward, ongoing engagement of pharmacists in health care operations will be a necessary feature to maximize health care value.

## 1. Introduction

Population health management is a data-driven process wherein opportunities are identified to improve the quality of health care delivered and thereby, promote better health outcomes for patients [1,2,3]. Data, including patient demographics, medical claims, and prescription claims, are collected and analyzed from multiple health information technology resources. This information is used to develop integrated, patient-centered care focused on disease management and prevention for a broad population. Overall, population health management attempts to maximize health care value by improving quality while reducing cost (Value = Quality/Cost).

Accountable care organizations (ACOs) are health care provider-based organizations that actively pursue population health management within their networks. They take responsibility for meeting the health care needs of a defined population with the goal of attaining the ‘Triple Aim’–defined by the Institute of Healthcare Improvement (IHI) as improved patient care experience, better population health and reduced per capita healthcare costs [4]. Since pharmaceutical spending has contributed substantially to rising healthcare costs [5], integration of pharmacists into ACOs is one way to support the achievement of the Triple Aim [6]. More than half of ACOs currently employ or contract pharmacists to support their mission [7]. The pharmacists are responsible for identifying opportunities to improve the quality and outcomes of care while managing pharmacy expenditures at both the individual and population levels. Data from medical and prescription claims are used to develop quality improvement projects intended to provide patients access to safe, efficacious, and cost-effective medication therapies. Pharmacists specifically focus on efforts to improve medication adherence; optimize drug selection that improves clinical outcomes while reducing inappropriate utilization; minimize variations in prescribing patterns; and improve patient satisfaction. Importantly, pharmacists collaborate with both providers and patients to achieve the best medication-related outcomes.

This article describes how two pharmacists embedded in Partners For Kids (PFK), one of the country’s oldest and largest pediatric ACOs, have conducted population health management for the large, diverse patient population PFK serves in central and southeast Ohio (Figure 1). The two pharmacists are part of PFK’s Office of the Medical Director team. Pharmacist-led interventions, ranging from the development of clinical decision support to medication management, have enabled PFK to effectively implement quality improvement initiatives aimed at improving health outcomes in behavioral health and asthma management for its pediatric patients.

## 2. Partners for Kids

Established as a physician/hospital organization in 1994, PFK has a governing board that includes representatives from Nationwide Children’s Hospital (NCH; Columbus, OH, USA) and from primary and specialty practice groups throughout Ohio that participate in the PFK provider network. PFK has assumed full medical and financial responsibility for 330,000 low-income children in central and southeastern Ohio (Figure 2).

Through subcontracted arrangements with Ohio’s five Medicaid Managed Care Plans, PFK is paid an age- and gender-adjusted capitation fee for each child each month (PMPM) that covers medical, dental, vision, and pharmacy services, as well as administrative expenses. PFK is fully responsible for managing and reimbursing providers for care. The managed care organizations retain a percentage of the administrative capitation rate to provide claims processing, member relations, and standard insurance management functions (Figure 3). PFK is considered an intermediary organization by the Ohio Department of Insurance and is responsible for maintaining reserves for future claims and re-insurance. PFK is committed to creating and sustaining a dynamic partnership with NCH, affiliated healthcare providers and the contracted managed care organizations to ensure access to the highest quality care for its members via various data measuring projects and specialized programs.

PFK differs from other ACOs in one key way. ACOs are identified based on their provider, meaning they contract with providers and subsequently are responsible only for the patients seen by those specific providers. In contrast, PFK assumes care for all children enrolled in a Medicaid managed care plan in its 34-county service region, even if the patient sees a provider not affiliated with PFK. Therefore, the primary goal is keeping the entire population as healthy as possible through population health management, with accrued savings reinvested into community health initiatives.

## 3. Quality Improvement

PFK actively engages with nearly 40, out of over 70, affiliated community practices, as well as NCH’s primary care network on quality improvement (QI) initiatives. QI specialists (QIS), clinical subject experts, and pharmacists work together closely to support primary care practices in developing and implementing QI projects across a spectrum of key issues facing pediatric patients. The QIS are largely responsible for training practice providers and office staff in the IHI Model for Improvement methodology and shepherding practice QI teams through identifying a project focus and aim, building a key driver diagram, developing and testing interventions, and tracking relevant process and outcome measures. This QI journey unfolds differently for each practice, as each QIS helps customize projects to meet the needs of each practice and/or its patients. For several QI projects focused on pediatric conditions with significant drug utilization, pharmacists play a critical role in optimizing prescribing and access to therapeutic options through interventions which are described below. PFK annually administers a survey to community providers affiliated with the ACO to qualitatively assess the impact of quality improvement projects on practice operations and patient-related health outcomes, as well as the usefulness of pharmacy resources for the providers. Responses to these questions are at an all-time high, with over 80% of participants responding favorably.

### 3.1. Prescribing Resources

PFK pharmacists develop prescribing resources, which include prescribing guidelines and a preferred drug list (PDL) that are easily accessible on the PFK website (www.PartnersForKids.org/Resources/). Prescribing guidelines for common conditions treated in the primary care setting assist providers in identifying the preferred medications for pediatric patients as defined by their clinical efficacy, safety, and cost-effectiveness. PFK pharmacists develop the prescribing guidelines using evidence-informed clinical guidelines and expert opinion (from NCH subject matter experts), where evidence is lacking. Formulary coverage and medication cost are also considered in the development of guidelines. Furthermore, PFK pharmacists work with the Medicaid Managed Care Plans to ensure that recommended medications are covered on their PDLs or formularies, ensuring less administrative burden for providers and less disruption for patients. Prescribing guidelines exist for acne, acute otitis externa and otitis media, behavioral health, head lice, gastroesophageal reflex, and outpatient antimicrobials, and guidelines for asthma and reproductive health are currently in development. In addition to prescribing guidelines, PFK maintains a PDL that is an abridged version of the five Medicaid Managed Care Plans and Ohio’s Fee-For-Service Medicaid’s PDLs. This PDL serves as a single reference for providers to quickly identify medication formulary coverage for patients regardless of insurance coverage. Anecdotally, through the annual PFK provider survey, community providers have found the prescribing guidelines and academic detailing provided by PFK pharmacists to be highly instructive for their practices. Additional topics for future guidelines have been identified through these surveys.

### 3.2. Education

PFK pharmacists educate community providers by disseminating prescribing resources and academic detailing. Electronic dissemination of prescribing resources is completed via the PFK website, monthly newsletters, quarterly webinars, and a free provider mobile app. Hard copies of prescribing resources are also distributed to community pediatricians by members of PFK and NCH’s outreach teams. PFK pharmacists conduct academic detailing at community pediatricians’ offices or during department section meetings. In addition to PFK’s prescribing resources, PFK pharmacists utilize and distribute practice or provider-specific data reports to support academic detailing efforts.

### 3.3. Clinical Decision Support

PFK pharmacists work with information technology specialists to develop clinical decision support. Specifically, they assist providers with preferred medication options within their practice-specific electronic medical record, including the creation or modification of alerts, favorites, and order sets. Clinical decision support provides timely information, at the point of care, to help inform decisions about a patient’s care and therapy options. Clinical decision support can effectively improve patient outcomes and lead to higher quality health care [8].

### 3.4. Medication Management

Using claims data, the two PFK pharmacists are able to identify opportunities for both PFK and NCH pharmacists to provide medication management. PFK pharmacists provide disease-specific medication management recommendations by working with community pediatricians using patient-level data reports provided by PFK. In addition, PFK pharmacists, NCH staff pharmacists at outpatient pharmacies, and NCH clinical pharmacists integrated into ambulatory clinics collaborate together through both comprehensive medication management (CMM) and disease-specific medication management to achieve the best medication-related outcomes for the PFK population.

CMM is a critical service provided by NCH clinical pharmacists integrated into NCH ambulatory clinics. These include primary care clinics and specialty clinics, such as adolescent medicine’s medication assisted treatment for addition clinic, Family AIDS Clinic and Educational Services program, complex care, endocrine, gastroenterology, neurology, pulmonary, and rheumatology clinics. NCH clinical pharmacists provide CMM as part of the medication reconciliation process. Common interventions that are employed include compliance assessments, medication reconciliation completions, medication education to both providers and patients, therapeutic adjustment recommendations, alternative therapy recommendations, and immunization recommendations. Over the past couple of years, NCH ambulatory clinical pharmacists have adopted a standardized documentation workflow within the electronic health record to track pharmacists’ interventions [9]. In addition, Ohio legislation expanded pharmacists’ services, allowing multiple pharmacists to have a collaborative practice agreement (CPA) with multiple physicians to manage drug therapy for multiple patients [10]. These two significant achievements position NCH ambulatory clinical pharmacists to explore future CPAs as a way to further enhance their roles in CMM.

Disease-specific medication management is another productive approach when an opportunity is identified—i.e., due to increasing medication costs or changes in insurance formulation—and pharmacists throughout the health system are available to implement a plan. PFK pharmacists often first identify such an opportunity, and then, in collaboration with the NCH clinical pharmacy team, devise and recommend alternative cost-effective and clinically-appropriate therapies when possible. With the integration of NCH clinical pharmacists in NCH ambulatory clinics, these initiatives can be executed easily with the medical team and patient/family.

Another example of collaboration is medication compliance assessments. Compliance assessment is a service that occurs while filling a prescription in NCH outpatient pharmacy with counseling at the time of dispensing, during medication reconciliation in an NCH ambulatory clinic by an NCH ambulatory clinical pharmacist, or through the use of claims data available to PFK pharmacists to assess medication adherence. Identifying patients’ adherence or non-adherence to their medications allows providers to work with high-risk children and their caregivers to overcome barriers to medication non-adherence, thereby resulting in better outcomes. For example, in the case of a patient with asthma, that type of intervention can translate into avoided ED visits or hospitalizations. With this multi-pronged approach, PFK and NCH pharmacists directly impact the care of patients and help keep them as healthy as possible.

## 4. Active Quality Improvement Initiatives with Pharmacist Involvement

As previously mentioned, pharmacist-led interventions, ranging from the development of clinical decision support to medication management, have enabled PFK to effectively implement quality improvement initiatives. Described below are quality improvement initiatives focused on behavioral health and asthma, the most common chronic conditions for pediatric patients.

### 4.1. Behavioral Health: Attention Deficit Hyperactivity Disorder

The objective of the Attention Deficit Hyperactivity Disorder (ADHD) preferred prescribing program was to optimize quality and cost-effective ADHD medication prescribing by primary care providers. In 2011, national survey data reported 14.2% of children in Ohio were diagnosed with ADHD by a health care provider, over half of whom took an ADHD medication [11]. ADHD accounts for nearly a quarter of PFK’s annual prescription drug expenditures. PFK pharmacists, in collaboration with expert pediatric psychiatrists, updated and published Prescribing Guidelines for Behavioral Health [12]. This tool utilizes evidence-informed clinical guidelines, cost information, and expert consultation to assist providers with timely and effective treatment for children with ADHD. Medications were designated as preferred based on clinical efficacy, safety, and cost-effectiveness.

PFK partnered with community practices as well as NCH’s primary care network (PCN) in 2017 to increase the prescribing rate of preferred ADHD medications. PFK pharmacists led the overall project, implementing interventions, such as sharing prescriber-specific data and feedback, presenting guidelines and data to attending physicians at section meetings and residents at monthly meetings, and developing ADHD-specific medication management interventions to review patient medications for appropriateness, effectiveness, and safety. Results showed a 3% increase in network-level prescribing rates of preferred ADHD medications in 2017 compared to 2016. In 2017, affiliated community practices were eligible for a physician incentive plan administered by PFK, in which rates of prescribing preferred ADHD medications was included as a measure. For a practice to receive incentive payments, the entire network must demonstrate cost savings through higher rates of ADHD preferred prescribing. During 2017, the network achieved significant savings, resulting in individual payouts for practices based on their patient volume.

Overall, these multidisciplinary QI interventions resulted in sustained improvements in prescribing rates of preferred ADHD medications, and as a result, have been able to realize cost savings without decreasing the quality of care. The PFK pharmacists actively continue to advance this initiative by working with the multidisciplinary team to identify and test new interventions, such as incorporating decision-support tools into the electronic health record system and advancing pharmacist involvement in medication management.

### 4.2. Behavioral Health: Project ECHO

In addition to pharmacists’ involvement in ADHD preferred prescribing, an NCH pharmacist participates in a PFK-led learning collaborative—Project ECHO. Project ECHO (Extension of Community Health Outcomes) is an innovative model that utilizes a ‘hub and spoke’ format to facilitate learning, wherein primary care providers at various distant sites (spokes) can interact with a multi-disciplinary expert team located at NCH (hub). This model has been utilized across various specialties and shown to have a significant impact on patient outcomes as well as participant efficacy [13]. PFK launched the first pediatric behavioral health Project ECHO in the state of Ohio in collaboration with NCH in 2019. As part of the hub team, the pharmacist plays an integral role as a content expert for psychopharmacology and related topics in the context of treatment of psychiatric disorders in the primary care settings. The expected impact of Project ECHO is a reduction in psychotropic polypharmacy and an overall improvement in the community behavioral health outcomes. Through this type of educational activity, pharmacists can maximize its potential impact by reaching a large group of pediatric providers.

### 4.3. Asthma

Asthma is one of the most common diseases affecting children, and medication management is a critical component in the care of asthma patients [14]. PFK has a QI project focused on the optimization of asthma management, with the specific aim to reduce asthma-related emergency department (ED) and inpatient visits by ensuring that patients with asthma are appropriately diagnosed, evaluated, and managed by pediatricians. Key interventions that are pursued by practices include (1) use of asthma control tests (ACT) to accurately assess a patient’s disease control and guide the need for changes in medication management; (2) distribution of asthma action plans (AAPs) to help patients appropriately self-manage symptoms; (3) workflow changes to support routine 6-month follow-up visits; and (4) strategies to improve asthma medication ratios (AMRs) for patients with more severe asthma [14].

PFK pharmacists have been a valuable resource in helping practices successfully adopt many of these interventions. For example, they provide best practice clinical guidelines for medication management through PFK’s PDL and academic detailing. Furthermore, as previously mentioned, a prescribing guideline for asthma is in development to assist providers in optimizing medication therapy based on ACT scores and also develop appropriate AAPs. PFK pharmacists also successfully advanced the use of AMR as a tool to track provider adherence to recommended asthma medication prescribing as well as patient adherence to prescribed medications. The AMR, defined as the units of controller medications prescribed for asthma over the units of controller and rescue medications, has been shown to correlate with good disease control and lower rates of ED visits for patients with persistent asthma [15]. Practices engaged in asthma QI were provided a list of patients with perceived poor disease control, defined by a low AMR and calculated using PFK pharmacy claims data so that they could develop individualized plans to improve their disease management. The PFK pharmacists also provided additional training on medication dispensing and administration to respiratory therapists and providers in practices. These on-site educational sessions have been well-received by primary care providers, and there is a growing interest to expand pharmacist-driven medication training to other disease states. Through the collaborative efforts of PFK QIS, pharmacists and primary care practices, participating practices have improved their management of asthma, and PFK patients have had better asthma-related outcomes.

## 5. Future Quality Improvement Initiatives with Pharmacist Involvement

Given how successful pharmacists have been in supporting key PFK priorities in ADHD and asthma, there is an opportunity to integrate pharmacy services in other areas, such as antimicrobial stewardship, reproductive health, and high cost or specialty medications. For example, inappropriate antibiotic prescribing for viral upper respiratory tract infections in children is an ongoing challenge, with substantial variation in prescribing patterns among our diverse network of providers. PFK pharmacists, partnering with their colleagues in NCH, are leading the way forward to build a community-facing/directed antimicrobial stewardship program. Currently, the team is in the process of developing and piloting interventions, such as patient-directed education on antibiotic avoidance, use of provider feedback to change prescribing practices, and education/training on interpreting local antibiograms to guide appropriate drug selection when indicated.

For primary care providers who see adolescent and young adult females in their clinics, reproductive health counseling and planning can be a challenging topic to navigate. The PFK QI team, in conjunction with pharmacists, is in the early stages of building a reproductive health QI project focused on expanding access to and education on contraception to sexually active female patients. The global aim of the QI project is to ensure that adolescents are receiving optimal reproductive and sexual health care at well-child exams. In addition to tracking the proportion of female adolescents in the PFK network with access to contraceptive planning, with a focus on long-acting reversible contraceptives, the project also monitors medication adherence by the patient, using PFK pharmacy and medical claims data. Pharmacists will be a resource to guide providers through which contraceptive plan is most appropriate in which setting, advise on anticipated side effects and symptom management, share changes in drug coverage in plan formularies, and assess medication adherence.

High cost or specialty medications are driving nearly 50% of total drug spend in health care today [16]. Strategies to ensure appropriate selection and utilization of medications are important to mitigate the risk of unnecessary spend on these agents. Equally important is optimizing adherence to these medications and analyzing total costs of care (medical and pharmacy), as these medications, when optimized, have the potential to improve quality of care for patients. Additionally, numerous specialty medications have been approved for the treatment of pediatric disease states in recent years. PFK pharmacists will facilitate the development of prescribing guidelines, including criteria for initiation and discontinuation of therapy. Outcomes research on the total cost of care will be important to inform PFK about the effectiveness of select specialty medications and to assess the quality of care delivery from a health system-owned specialty pharmacy.

## 6. Conclusions

Pharmacists at PFK play an essential role in improving children’s health outcomes through population health management strategies. PFK pharmacists conduct academic detailing, design medication management, assist with clinical decision support, and collaborate with healthcare providers to ensure clinically-appropriate and cost-effective medications are accessible to PFK patients. Through analysis of medical and prescription claims data, development of prescribing resources for formulary management and common pediatric conditions, and implementation of quality improvement initiatives at the practice-level, PFK pharmacists have identified critical opportunities and provided valuable input to improve the direction of care delivered for central and southeastern Ohio children. In the future, PFK pharmacists will continue to use these methods and others to pursue additional opportunities to improve pediatric population health. The successful integration of pharmacists within PFK can serve as a model for other ACOs looking for innovative ways to promote value-based health care while containing health care costs.

## Figures and Tables

**Figure 1 children-06-00082-f001:**
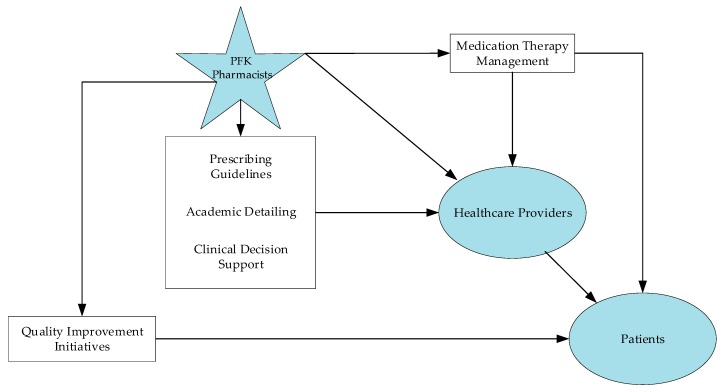
Diagram of population health management strategies employed by pharmacists working within a pediatric accountable care organizations (ACO) to help deliver high-value health care to its patients.

**Figure 2 children-06-00082-f002:**
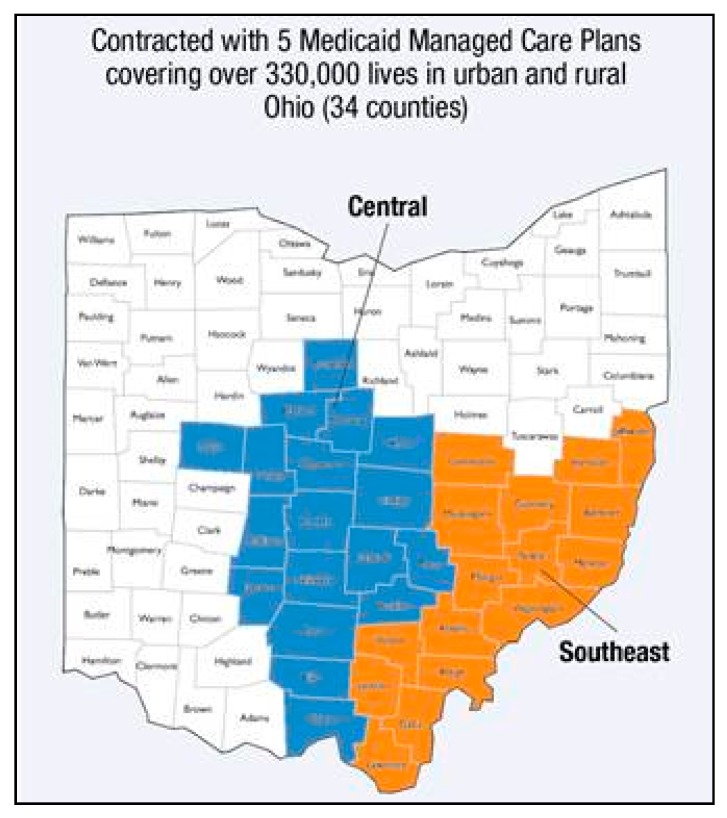
Map of Ohio with the highlighted region indicating which county Partners For Kids (PFK) patients live.

**Figure 3 children-06-00082-f003:**
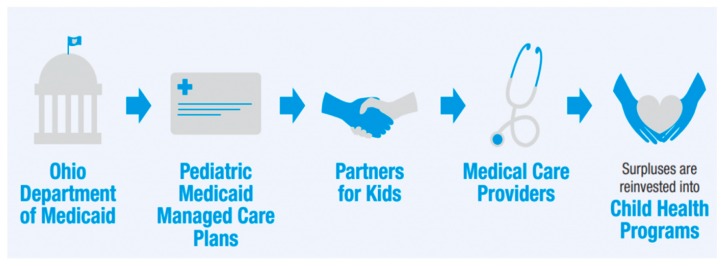
Partners for Kids flow of funds.

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
