# Peer review of "Pharmacist Involvement in Population Health Management for a Pediatric Managed Medicaid Accountable Care Organization"

_children, 2019, doi:10.3390/children6070082_

Reviewer 1 Report

Thank you for the revision.

Reviewer 2 Report

The authors have adjusted the manuscript in accordance with the recommendations for harmonizing with the Special Issue. The additional information and synthesis is very helpful for providing context and results for this very important practice change. Well done!

This manuscript is a resubmission of an earlier submission. The following is a list of the peer review reports and author responses from that submission.

Round  1

Reviewer 1 Report

Thank you for the submission.  The work is quite interesting and well-written. I hope that it will be of interest to our Journal’s readers.

The one improvement I would suggest is the inclusion of some details on the structure and working of PFK with respect to pharmacy – in particular, how many pharmacists are employed by PFK and do they work centrally as a team etc.?  The manuscript uses the term “PFK pharmacists” but it is not clear whether this is referring to pharmacists working centrally for PFK or whether this includes community pharmacists receiving fees for services provided on behalf of PFK e.g. who provides the academic detailing of prescribers? How does the prescribing guidelines development process work? Is there external and independent review of the guidelines?

Reviewer 2 Report

The authors present a very interesting description of a long-standing population health management program that incorporates pharmacists to provide medication management and other medication-related activities focused on patients with attention-deficit hyperactivity disorder and asthma. They mention that the program has been successful for meeting so called IHI triple aim. The following suggestions would strengthen the paper and better integrate it within the special issue:

Can the authors can include data on therapeutic outcomes, program cost reductions, and/or increases in patient / provider satisfaction achieved because on incorporation of PFK and clinical pharmacists? In addition, data on the use of collaborative practice agreements (CPA) between pediatricians/primary care providers and clinical pharmacists in network ambulatory care sites or level of pharmacist board certification would help to integrate this paper into a major theme of the special issue. 

The special issue promotes comprehensive medication management (CMM) through CPA as a best practice for direct care provided by pharmacists for children with special healthcare needs (CSHCN). Medicare Part D uses the term Medication Therapy Management (MTM) specifically for adult patients with chronic diseases, but has no practice component for children. MTM practice description at lines 144-147 (compliance assessment, medication reconciliation, medication adjustments, etc.) are also components of CMM. The difference between CMM and MTM is the scope of the problems to be identified and managed. To avoid confusion between CMM and MTM, can the authors support harmonizing the terminology describing pharmacist practice in the manuscript without compromising their service description? Consider changing the heading at line 137 to 'Medication Management.' Can we refer to MTM processes as disease-specific and CMM as a whole-person approach because of multiple conditions?

Another focus of the special issue is on children with special healthcare needs with medical complexity (CSHCN-CMC). Do you have any examples of how these patients are managed. For example, a patient may have asthma, GERD, and ADHD. How would pharmacist care be provided to patients with multiple co-morbidities?

Overall, the manuscript is very readable and the descriptions thorough. Nice work.